# Crystallographic Calculations and First-Principles Calculations of Heterogeneous Nucleation Potency of γ-Fe on La_2_O_2_S Particles

**DOI:** 10.3390/ma15041374

**Published:** 2022-02-12

**Authors:** Yin Zhou, Yunping Ji, Yiming Li, Jianbo Qi, Haohao Xin, Huiping Ren

**Affiliations:** 1School of Materials and Metallurgy (School of Rare Earth), Inner Mongolia University of Science and Technology, Baotou 014010, China; zhouyin0814@163.com (Y.Z.); qjbqjb1980@126.com (J.Q.); xinhh1996@163.com (H.X.); 2Inner Mongolia Autonomous Region Key Laboratory of Advanced Metal Materials, Baotou 014010, China; liyiming79@sina.com (Y.L.); renhuiping@sina.com (H.R.); 3Collaborative Innovation Center of Integrated Exploitation of Bayan Obo Multi-Metal Resources, Inner Mongolia University of Science and Technology, Baotou 014010, China

**Keywords:** La_2_O_2_S, γ-austenite, heterogeneous nucleation, E2EM, first-principles calculations

## Abstract

Rare earth (RE) inclusions with high melting points as heterogeneous nucleation in liquid steel have stimulated many recent studies. Evaluating the potency of RE inclusions as heterogeneous nucleation sites of the primary phase is still a challenge. In this work, the edge-to-edge matching (E2EM) model was employed to calculate the atomic matching mismatch and predict the orientation relationship between La_2_O_2_S and γ-Fe from a crystallographic point of view. A rough orientation relationship (OR) was predicted with the minimum values of $f_{r}=9.43\%$ and $f_{d}=20.72\%$ as follows: ${\left[2\bar{1}\bar{1}0\right]}_{{\mathrm{La}}_{2}O_{2}S}$∥${\left[100\right]}_{\gamma -\mathrm{Fe}}$ and ${\left(000\bar{3}\right)}_{{\mathrm{La}}_{2}O_{2}S}$∥${\left(00\bar{2}\right)}_{\gamma -\mathrm{Fe}}$. The interface energy and bonding characteristics between La_2_O_2_S and γ-Fe were calculated on the atomic scale based on a crystallographic study using the first-principles calculation method. The calculations of the interface energy showed that the S-terminated and La(S)-terminated interface structures were more stable. The results of difference charge density, electron localization function (ELF), the Bader charges and the partial density of states (PDOS) study indicated that the La(S)-terminated interface possessed metallic bonds and ionic bonds, and the S-terminated interface exhibited metallic bond and covalent bond characteristics. This work addressed the stability and the characteristics of the La_2_O_2_S/γ-Fe interface structure from the standpoint of crystallography and energetics, which provides an effective theoretical support to the study the heterogeneous nucleation mechanism. As a result, La_2_O_2_S particles are not an effective heterogeneous nucleation site for the γ-Fe matrix from crystallography and energetics points of view.

## 1. Introduction

Previous studies showed that adding the rare earth element La/Ce into steels resulted either in a decrease in the proportion of the columnar grains or the average grain size of the equiaxed grains or both in the case of as-cast austenitic [1,2,3], ferritic [4,5,6] and other engineering steels [7,8]. The most generally accepted effect mechanisms of La/Ce are the heterogeneous nucleation effect of the primary δ-ferrite/γ-austenite on La/Ce-containing inclusions with high melting points, including oxides, oxysulfides and sulfides [1,2,4,5,6,9], or/and the solute effects [2,10] attributed to the solute enrichment or depletion in the liquid phase ahead of the solid-liquid interface. In the past decades, the two-dimensional lattice disregistry model [11] has been widely adopted to evaluate the heterogeneous nucleation potency. A potency evaluation criterion based on the two-dimensional lattice disregistry model was proposed [11]: when the lattice disregistry value (*δ*-value) is below 6%, the potency is the most effective, and the grains can be significantly refined; when the *δ*-value is between 6% and 12%, the potency is moderately effective; when the *δ*-value is above 12%, the potency is poor, and the grains cannot be refined. Based on this model, researchers reported the heterogeneous nucleation potency of La/Ce oxides, oxysulfides and sulfides for the primary δ-ferrite and/or primary γ-austenite during steel solidification, examined using the two-dimensional disregistry model [2,6,12,13]. However, for a given La/Ce-containing inclusion phase and the primary δ-ferrite/γ-austenite, the *δ*-values they obtained were highly scattered. Relevant studies have been reviewed by the authors [14]. The heterogeneous nucleation potency of La/Ce oxides, oxysulfides and sulfides for the primary δ-ferrite/γ-austenite was inconsistent in relevant studies. The discrepancy might have resulted from different selection standards. Put another way, the selecting criteria of low index planes and directions for *δ*-value calculation were not defined based on the two-dimensional lattice disregistry model. Yu and co-workers [15] calculated the lattice disregistry between La_2_O_2_S and γ-austenite in an Fe-43Ni expansion alloy and reported the lowest *δ*-value of 5.42% between the (0001) crystal plane of La_2_O_2_S and the (100) crystal plane of γ-austenite. This implied a high heterogeneous nucleation potency of La_2_O_2_S for γ-austenite. However, there was no conclusive experimental evidence to verify it. Robert Tuttle’s work [16,17,18] showed that the addition of lanthanum can refine the solidification structure, but not all lanthanum inclusions act as heterogeneous nucleus of γ-austenite. The heterogeneous nucleation potency of La-containing inclusion phases for the primary γ-austenite needs further investigation.

Recently, the edge-to-edge matching (E2EM) model [19] originally developed by Zhang and Kelly was validated to be a powerful crystallography model in interpreting the heterogeneous nucleation potency in the grain refinement during solidification and predicting effective heterogeneous nucleation agents in Al, Mg and their alloys [20,21,22]. The major advantage of the E2EM model over the two-dimensional lattice disregistry model is that it focuses on actual atom matching rather than lattice matching. It can not only calculate the atomic matching between two phases but also predict the favorable orientation relationships (ORs) between the two phases (see refs. [18,19,20,22] for related details and calculation method). First-principles calculation based on density functional theory is a very effective method for studying alloy interfaces [23,24,25]. The density functional theory (DFT) method is widely applied to investigate the material interface of the magnesium matrix [23,26] and aluminum matrix [24]. Moreover, for the interface between compounds (such as LaAlO_3_ [27], La_2_O_3_ [25], Al_2_O_3_ [28], TiC [29] and TiN [30]) and the δ-ferrite/γ-austenite matrix, the DFT calculations are also employed to probe the interfacial characteristics and the mechanism of heterogeneous nucleation. In this paper, La_2_O_2_S, which is the typical oxysulfide, was selected. The present work aims to clarify the heterogeneous nucleation potency of γ-Fe on La_2_O_2_S particles, combining a crystallographic calculation using the edge-to-edge matching model with the first-principles calculations from an energetics point of view.

## 2. Calculations

### 2.1. Crystallographic Calculations Using the E2EM Model

To investigate the heterogeneous nucleation potency of La_2_O_2_S for γ-austenite, the E2EM model was adopted for crystallographic calculations. The interatomic spacing misfit (f_r_) along the close-packed (CP) or nearly CP matching rows, as well as the interplanar spacing mismatch (f_d_) of the CP or nearly CP planes between La_2_O_2_S and γ-austenite, were obtained based on the crystal structures, lattice parameters and atomic positions in the two phases. Table 1 presents the crystal structure, space group, lattice parameters and atomic positions of γ-austenite (γ-Fe) and La_2_O_2_S.

The γ-Fe phase possesses a face-centered cubic (FCC) lattice structure with four atoms in each unit cell. By contrast, the La_2_O_2_S phase has a primitive hexagonal structure, and each unit cell contains two lanthanum atoms, two oxygen atoms and one sulfur atom. The γ-Fe lattice parameters were [31]: a = 0.36544 nm; space group = $\mathrm{Fm}\bar{3}m$. The La_2_O_2_S lattice parameters were [32]: a = 0.4035 nm; c = 0.6914 nm; space group = $P\bar{3}m1$.

### 2.2. First-Principles Methodology

Effective heterogeneous nucleation must satisfy the crystallographic match and interface stability between the nucleating agent and the nucleating phase. The plane-wave, pseudo-potential density functional theory (DFT) method implemented in the VASP package [33] was used to investigate the stability of La_2_O_2_S/γ-Fe interface. The interaction between ion and valence electrons was described by the projector augmented wave (PAW) [34]. The exchange correlation energy between electrons was employed using the local density approximation (LDA) [35] and the generalized gradient approximation (GGA) with the Perdew-Burke-Ernzerhof (PBE) functional [36]. Due to the presence of La atoms in the system, strong correlation effects had to be considered. Thus, the LDA+U/GGA+U functions were adopted as the exchange correlation functions. Moreover, the effective value for U was 6 eV [37]. Since Fe atoms are magnetic, the spin-polarization method was employed. The Kohn-Sham equation was solved with self-consistent field (SCF), and the electronic minimization process was conducted to reach the ground state [36]. 

In order to verify the methodology proposed in this work, the bulk properties of La_2_O_2_S and γ-Fe were investigated, and the results were compared with published experimental data. To estimate the necessary layer numbers of bulk-like slabs N used in the subsequent interface systems, the surface properties of slabs with different layer numbers were studied. The interface properties of different atomic terminals, including adhesion work, interface energy and interface bonding properties, were calculated to further investigate the interface stability.

In the structural relaxation, the plane-wave cutoff E_cut_ was 450 eV. In the γ-Fe and La_2_O_2_S bulk phase optimization, the Brillouin zone integral was divided by 9 × 9 × 9 and 10 × 10 × 5 Monkhorst-Pack k-point grids (centered on the gamma point), respectively, which was proven to be sufficient in our convergence test. The k-points sampling grids of the surface and interface were generated in the Monkhorst-Pack method of 0.04 grid density by VASPKIT [38]. A vacuum gap of 15 Å was inserted to eliminate periodical influence in the Z direction. According to the previous DFT calculations regarding other condensed matter interfaces [24,28,39], a vacuum depth of 15 Å is thick enough to generate accurate results. The energy changes in the structural optimization process converged to 1.0 × 10^−4^ eV/atom, and the residual Hellmann–Feynman force on each relaxed atom reached 0.03 eV/Å [27,28,29].

## 3. Results and Discussion

### 3.1. Nucleation Crystallography of γ-Fe on La_2_O_2_S Particles

Based on crystal structures, lattice parameters and atom positions, the E2EM calculations of the CP or nearly CP rows and the CP or nearly CP planes of γ-Fe and La_2_O_2_S were obtained, as shown in Table 2. It can be seen that the smaller the atomic spacing, the denser the arrangement; the larger the face spacing, the denser the arrangement. The CP atomic rows were straight, pseudo-straight or zigzag, denoted by SS, PS and ZZ, respectively. Then, based on the identified CP rows and CP planes, the values of f_r_ and f_d_ between La_2_O_2_S and γ-Fe were calculated, as listed in Table 3. It can be seen that f_r_ was 9.43%, and the minimum f_d_ was 4.58%.

Because the matching rows must be on the corresponding matching planes, a favorable orientation relationship (OR) between La_2_O_2_S and γ-Fe could be predicted based on the matching directions and matching planes. The predicted rough OR between La_2_O_2_S and γ-Fe could be expressed as ${\left[2\bar{1}\bar{1}0\right]}_{{\mathrm{La}}_{2}O_{2}S}$∥${\left[100\right]}_{\gamma -\mathrm{Fe}}$ and ${\left(000\bar{3}\right)}_{{\mathrm{La}}_{2}O_{2}S}$∥${\left(00\bar{2}\right)}_{\gamma -\mathrm{Fe}}$, and the values of f_r_ and f_d_ were 9.43% and 20.72%, respectively. Based on this orientation relationship, the properties of the heterogeneous interface between La_2_O_2_S and γ-Fe were calculated by the first principles. The effectiveness of La_2_O_2_S as the heterogeneous nucleation site of γ-Fe was discussed from the point of view of energy.

### 3.2. Bulk and Surface Properties

#### 3.2.1. Bulk Calculations

Table 4 and Table 5 present the optimized γ-Fe and La_2_O_2_S lattice constants (a and c), bulk module (B_0_) and phase equilibrium volume per atom (V_0_) by LDA and GGA-PBE function. For γ-Fe bulk, the calculated constants by GGA-PBE function were a = b = c = 3.636 Å, 0.5% below the experimental values [31] a = b = c = 3.654 Å. For La_2_O_2_S phase, the calculated lattice constants were a = b = 4.134 Å, and c = 7.073 Å. They are roughly consistent with the experimental values [32] a = b = 4.035 Å and c = 6.914 Å. The results of GGA-PBE functional also agreed well with the reported calculations in Table 4 and Table 5. This ensured the reliability of the presented calculations.

#### 3.2.2. Surface Convergence Test

In view of the lattice match of the interface, γ-Fe$\left(00\bar{2}\right)$ and La_2_O_2_S$\left(000\bar{3}\right)$ surfaces were selected to construct their interface. The surface model should be thick enough to ensure the bulk-like character interiors so as to obtain accurate surface properties. It was, therefore, necessary to perform a convergence test on the slabs to construct the interface calculations. The 5-layer, 7-layer, 9-layer and 11-layer slabs were respectively modeled for the convergence test of γ-Fe$\left(00\bar{2}\right)$ through the method presented in refs. [25,28,37]. The surface energy can be expressed by [30]:(1)$$E_{\mathrm{surf}}=\left(E_{\mathrm{slab}}-\left(N_{\mathrm{slab}}/N_{\mathrm{bulk}}\right)E_{\mathrm{bulk}}\right)/2A,$$
where $E_{\mathrm{slab}}$ is the total energy of the slab; $E_{\mathrm{bulk}}$ is the total energy of bulk after energy optimization; A is the corresponding surface area; and $N_{\mathrm{slab}}$ and $N_{\mathrm{bulk}}$ are the number of atoms in the slab and bulk structure, respectively. Table 6 gives the surface energy of γ-Fe stoichiometric slab corresponding to the number of atom layers. Obviously, the γ-Fe$\left(00\bar{2}\right)$ surface energy converged to 2.18 J/m^2^ when the surface model contained more than five atomic layers. This figure is close to the experimental value 1.95 J/m^2^ reported in ref. [39] Therefore, the five-layer γ-Fe$\left(00\bar{2}\right)$ surface slab was adopted to ensure the bulk-like interior in the following calculations.

For the La_2_O_2_S$\left(000\bar{3}\right)$ slab, there were three different kinds of surface terminations (La-, O- and S-terminated). Both La- and O-terminated surfaces had two nonequivalent surfaces with different sub-layer atoms. The sub-layer atoms were O or S for the La-terminated surface and La or O for O-terminated. Symmetric slabs were adopted in the surface convergence tests to eliminate the spurious dipole effects. Therefore, there were five possible types of slab surface, as shown in Figure 1.

Table 7, Table 8 and Table 9 show the La_2_O_2_S$\left(000\bar{3}\right)$ surface relaxations of the different atomic layers with different terminals before and after the change of the atomic layer spacing accounted for the percentage of the interlayer spacing of the bulk. It can be seen that the effects of atomic relaxations were mainly localized within the top three atomic layers. With the increasing atomic layers, the slab interiors became gradually bulk-like in nature. In the case of the 14-layer, 13-layer, 15-layer, 12-layer and 11-layer structure, respectively, the La(O)-terminated, La(S)-terminated, O(La)-terminated, O(O)-terminated and S-terminated surface interlayer relaxations were well converged. Therefore, the corresponding slabs were applied to simulate their interface structure. The results in Table 7, Table 8 and Table 9 demonstrate that the outermost and second interlayer relaxations for S-terminated surface were smallest among the five types of surface, followed by the La(S)-terminated surface. It was observed that the outermost interlayer distances of S-terminated and La(O)-terminated surfaces with eleven and thirteen atomic layers were reduced/enlarged by 1.037% and 5.663%, and the second interlayer distances were enlarged/reduced by 2.080% and 2.121%, respectively. Thus, the S-terminated and La(O)-terminated surfaces tended to be more stable, with no surface reconstruction.

#### 3.2.3. Surface Stability of La_2_O_2_S$\left(000\bar{3}\right)$

The atomic species of La_2_O_2_S surface have significant impact on the interfacial stability. Based on the surface convergence test result, the stability of different terminal surfaces was investigated. For surface energy of La_2_O_2_S, characterized by non-stoichiometry, La, O and S chemical potentials needed to be considered comprehensively to accurately determine the stability in the experimental environment. The surface energy of La_2_O_2_S was calculated by Equation (2) [23,25,26,27,28,42]:(2)$$\delta_{{\text{La}}_{\text{2}}{\text{O}}_{\text{2}}\text{S}(000\bar{3})}=\left[E_{\text{slab}}-N_{\text{La}}\mu_{\text{La}}-N_{\text{O}}\mu_{\text{O}}-N_{\text{S}}\mu_{\text{S}}+PV-TS\right]/2A,$$
where $E_{\mathrm{slab}}$ is total energy of the corresponding slab, $N_{i}$ is the number of atomic species i (i = La, O and S) in the La_2_O_2_S slab, $\mu_{i}$ is chemical potentials of atomic species i (i = La, O and S) at ground state and A is surface area. The calculations were carried out at 0 K and condensed-matter states, so the PV and TS terms can be neglected [28].

Combining the chemical potentials of elements La, O and S of the La_2_O_2_S slab in equilibrium with those of bulk La_2_O_2_S, the relationship between the chemical potential of La_2_O_2_S bulk $\mu_{{\mathrm{La}}_{2}O_{2}S}^{\mathrm{bulk}}$, its 0 K formation heat ($\Delta H_{f}^{0}\left({\mathrm{La}}_{2}O_{2}S\right)$) and chemical potentials of the La bulk, O_2_ gas and S bulk ($\mu_{\mathrm{La}}^{\mathrm{bulk}}$, $\frac{1}{2}\mu_{O_{2}}^{\mathrm{gas}}$ and $\mu_{S}^{\mathrm{bulk}}$) are expressed by [43]:(3)$$\mu_{{\mathrm{La}}_{2}O_{2}S}^{\mathrm{bulk}}=2\mu_{\mathrm{La}}+2\mu_{O}+\mu_{S},$$
(4)$$\mu_{{\mathrm{La}}_{2}O_{2}S}^{\mathrm{bulk}}=2\mu_{\mathrm{La}}^{\mathrm{bulk}}+\mu_{O_{2}}^{\mathrm{gas}}+\mu_{S}^{\mathrm{bulk}}+\Delta H_{f}^{0}\left({\mathrm{La}}_{2}O_{2}S\right),$$
where $\mu_{i}$ is chemical potentials of atomic species i (i = La, O and S), the actual value of which is located in the surface within a certain range and is determined by experimental conditions. The upper bound of $\mu_{i}$ (i = La, O and S) at the surface is the corresponding potential in pure La bulk, gas phase of O_2_ and pure S bulk: $\mu_{\mathrm{La}}\leq \mu_{\mathrm{La}}^{\mathrm{bulk}}$, $\mu_{O}\leq \frac{1}{2}\mu_{O_{2}}^{\mathrm{gas}}$ and $\mu_{S}\leq \mu_{S}^{\mathrm{bulk}}$, which is the basis for forming stable slab structure. The range of $\mu_{\mathrm{La}}-\mu_{\mathrm{La}}^{\mathrm{bulk}}$ was obtained:(5)$$\Delta \mu_{\mathrm{La}}=\mu_{\mathrm{La}}-\mu_{\mathrm{La}}^{\mathrm{bulk}}\leq 0,$$
(6)$$\Delta \mu_{O}=\mu_{O}-\frac{1}{2}\mu_{O_{2}}^{\mathrm{gas}}\leq 0,$$
(7)$$\Delta \mu_{S}=\mu_{S}-\mu_{S}^{\mathrm{bulk}}\leq 0,$$

Based on Equations (3), (5) and (7), the surface energy (Equation (2)) can be rewritten as:(8)$$\delta_{{\text{La}}_{\text{2}}{\text{O}}_{\text{2}}\text{S}(000\bar{3})}=\left[\begin{array}{c} E_{\text{slab}}-\frac{N_{\text{O}}}{2}\mu_{{\text{La}}_{\text{2}}{\text{O}}_{\text{2}}\text{S}}^{\text{bulk}}+\left(N_{\text{O}}-N_{\text{La}}\right)\mu_{\text{La}}^{\text{bulk}}+\left(N_{\text{O}}-N_{\text{La}}\right)\Delta \mu_{\text{La}}\\ +\left(\frac{N_{\text{O}}}{2}-N_{\text{S}}\right)\mu_{\text{S}}^{\text{bulk}}+\left(\frac{N_{\text{O}}}{2}-N_{\text{S}}\right)\Delta \mu_{\text{S}} \end{array}\right]/2A,$$
where $\mu_{\mathrm{La}}^{\mathrm{bulk}}$ and $\mu_{S}^{\mathrm{bulk}}$ refers to the single atomic energy in the La and S bulk; $\mu_{{\mathrm{La}}_{2}O_{2}S}^{\mathrm{bulk}}$ is equal to the energy of La_2_O_2_S bulk per formula unit (f.u.); and $N_{i}$ (i = La, O and S) has the same meaning as in Equation (2). For La_2_O_2_S bulk as a heterogeneous nucleus, the actual solidification process was in a S-rich environment, so the S chemical potential was approximated to zero ($\Delta \mu_{S}=0$). The range of the $\Delta \mu_{\mathrm{La}}$ value combined with the formula available for La chemical potential (Equations (4)–(6)) can be derived as:(9)$$\Delta H_{f}^{0}\left({\mathrm{La}}_{2}O_{2}S\right)\leq \Delta \mu_{\mathrm{La}}\leq 0,$$

The formation heat $\Delta H_{f}^{0}\left({\mathrm{La}}_{2}O_{2}S\right)$ was calculated to be −5.12 eV. The range of $\Delta \mu_{\mathrm{La}}$ was, therefore, derived to be:(10)$$-5.12\;\mathrm{eV}\leq \Delta \mu_{\mathrm{La}}\leq 0\;\mathrm{eV},$$

Figure 2 shows La_2_O_2_S$\left(000\bar{3}\right)$ surface energy varying with the variation of the chemical potential of $\Delta \mu_{\mathrm{La}}$. Clearly, the surface energy of these surfaces was linearly correlated with $\Delta \mu_{\mathrm{La}}$. Generally, the lower the surface energy is, the more stable the surface structure is. In the whole range of La chemical potential, the surface energy of the La(S)-terminated surface was the lowest, and the values decreased with the increases of $\Delta \mu_{\mathrm{La}}$ in the range of −0.74 J/m^2^~−6.29 J/m^2^. By contrast, the surface energy of La(O)-terminated, O(La)-terminated and S-terminated surfaces remained constant at 4.79 J/m^2^, 0.68 J/m^2^ and 0.06 J/m^2^, respectively. It should be pointed out that the constant value was due to the zero value of ($N_{O}-N_{\mathrm{La}}$) in Equation (10). Therefore, the La(S)-terminated surface was the most stable one over the entire range of chemical potential.

### 3.3. Interface Properties between La_2_O_2_S and γ-Fe

#### 3.3.1. Structures of La_2_O_2_S/γ-Fe Interface

Based on the surface convergence test results and the orientation relationship predicted by edge-to-edge matching crystallography theory, five interface models between La_2_O_2_S and γ-Fe were constructed. They were denoted by La(O)-terminated, La(S)-terminated, O(La)-terminated, O(O)-terminated and S-terminated La_2_O_2_S/γ-Fe interfaces, as shown in Figure 3. A supercell geometry where a five-layer γ-Fe$\left(00\bar{2}\right)$ slab was placed under a corresponding, symmetric atomic terminals La_2_O_2_S$\left(000\bar{3}\right)$ slab was used to simulate the La_2_O_2_S/γ-Fe interface. For interfacing the close-packed planes, the slabs were rotated about an axis normal to the interface so as to align the close-packed directions, resulting in an orientation relationship of ${\left[2\bar{1}\bar{1}0\right]}_{{\mathrm{La}}_{2}O_{2}S}$∥${\left[100\right]}_{\gamma -\mathrm{Fe}}$ and ${\left(000\bar{3}\right)}_{{\mathrm{La}}_{2}O_{2}S}$∥${\left(00\bar{2}\right)}_{\gamma -\mathrm{Fe}}$. To accommodate the periodic boundary conditions inherent in a supercell calculation, the coherent interface approximation [44] was invoked, and the average of the lattice parameters for La_2_O_2_S and γ-Fe was used. The interface structures were fully relaxed in all dimensions by minimizing the Hellmann-Feynman force on every atom.

#### 3.3.2. Interface Stability

Binding strength of the interface can be predicted by the work of adhesion ($W_{\mathrm{ad}}$), which is defined as the reversible work required to separate the interface between A and B phase to form two free surfaces. The larger the $W_{\mathrm{ad}}$ is, the stronger the binding force of interface atoms is. The $W_{\mathrm{ad}}$ of the interface can be calculated by Equation (11) [23,24,25,26,27,28,30]:(11)$$W_{\mathrm{ad}}=\left(E_{{\mathrm{La}}_{2}O_{2}S}+E_{\gamma -\mathrm{Fe}}-E_{{\mathrm{La}}_{2}O_{2}S/\gamma -\mathrm{Fe}}\right)/A,$$
where $E_{{\mathrm{La}}_{2}O_{2}S/\gamma -\mathrm{Fe}}$ is the total energy of interface system; $E_{{\mathrm{La}}_{2}O_{2}S}$ and $E_{\gamma -\mathrm{Fe}}$ are the total energy of isolated La_2_O_2_S and γ-Fe slabs in the same interface system when one of the slabs is kept and other is replaced with vacuum, respectively; and A is the interface area. Generally, a more positive $W_{\mathrm{ad}}$ means a larger binding strength.

Table 10 presents the optimal $W_{\mathrm{ad}}$ for the relaxed geometries of the five La_2_O_2_S/γ-Fe interfaces. The $W_{\mathrm{ad}}$ of the five La_2_O_2_S/γ-Fe interfaces in descending order was: S-terminated > La(S)-terminated > La(O)-terminated > O(O)-terminated > O(La)-terminated. This indicates that the S-terminated interface had the largest binging strength, followed by the La(S)-terminated interface. This partly resulted from the bonding type between the interface atoms. In addition, the S-terminated and La(S)-terminated surface structures were more stable than others with a smaller surface energy. These demonstrate that it is more likely that the S-terminated and La(S)-terminated interfaces are maintained.

In addition to the work of adhesion ($W_{\mathrm{ad}}$), interface energy is another major indicator for measuring interface stability. In general, the lower the interface energy becomes, the more stable the interface structure is. The interface energy of La_2_O_2_S/γ-Fe was calculated from the viewpoint of thermodynamics, and the interface stability of the different terminated interface configurations was quantitatively analyzed. The results can provide theoretical data for the heterogeneous nucleation mechanism of La_2_O_2_S on the γ-Fe substrate. According to the calculation method provided in refs. [23,26,30], the interface energy can be derived as follows:(12)$$\gamma_{{\mathrm{La}}_{2}O_{2}S/\gamma -\mathrm{Fe}}=\left[\begin{array}{c} E_{{\mathrm{La}}_{2}O_{2}S/\gamma -\mathrm{Fe}}-\frac{N_{O}}{2}\mu_{{\mathrm{La}}_{2}O_{2}S}^{\mathrm{bulk}}+\left(N_{O}-N_{\mathrm{La}}\right)\mu_{\mathrm{La}}^{\mathrm{bulk}}+\left(N_{O}-N_{\mathrm{La}}\right)\Delta \mu_{\mathrm{La}}\\ +\left(\frac{N_{O}}{2}-N_{S}\right)\mu_{S}^{\mathrm{bulk}}-N_{\mathrm{Fe}}\mu_{\mathrm{Fe}}^{\mathrm{bulk}} \end{array}\right]/A-\delta_{{\mathrm{La}}_{2}O_{2}S}-\delta_{\gamma -\mathrm{Fe}},$$
where $E_{{\mathrm{La}}_{2}O_{2}S/\gamma -\mathrm{Fe}}$ is the total energy of interface system; $\mu_{\mathrm{Fe}}^{\mathrm{bulk}}$ is equal to the energy of γ-Fe bulk per formula unit (f.u.); $\delta_{{\mathrm{La}}_{2}O_{2}S}$ and $\delta_{\gamma -\mathrm{Fe}}$ denote the surface energy of La_2_O_2_S and γ-Fe free surfaces in the same interface system; and the other parameters are consistent with those in Equation (8).

Figure 4 shows the interface energy of La_2_O_2_S/γ-Fe interfaces with the increase of $\Delta \mu_{\mathrm{La}}$. The La_2_O_2_S/γ-Fe structures of S-terminated and La(S)-terminated interfaces, which have smaller interface energy, were more stable than others. It, therefore, can be concluded that S-terminated and La(S)-terminated interfaces are most likely to form. Within the range of $-5.12\;\mathrm{eV}\leq \Delta \mu_{\mathrm{La}}\leq -4.69\;\mathrm{eV}$, the interface energy of the S-terminated interface structure was smaller than that of the La(S)-terminated interface structure. This indicates that the S-terminated interface had a more stable structure. In the case of $-4.69\;\mathrm{eV}\leq \Delta \mu_{\mathrm{La}}\leq 0\;\mathrm{eV}$, the interface energy of S-terminated interface structure was larger than that of La(S)-terminated interface structure. Thus, the La(S)-terminated interface structure was more stable.

#### 3.3.3. Electronic Structure and Bonding

The bonding strength of the interface is closely related to the bonding type of the interface atoms, as well as the electronic structure of the interface. To better understand interface atomic bonding, the difference charge density, the electron localization function (ELF), the Bader charges and the partial density of states (PDOS) were employed. The electronic structure and bonding of the more stable S-terminated and La(S)-terminated interface structures were compared to explore the interface stability.

The difference charge density can be used to analyze the transfer of interatomic charge. The charge accumulation and depletion regions were ascertained. The charge density difference was calculated by the equation below [28,30]:(13)$$\Delta \rho =\rho_{\mathrm{int}}-\rho_{{\mathrm{La}}_{2}O_{2}S}-\rho_{\gamma -\mathrm{Fe}},$$
where $\rho_{\mathrm{int}}$ is the total charge density of La_2_O_2_S/γ-Fe interface system; and $\rho_{{\mathrm{La}}_{2}O_{2}S}$ and $\rho_{\gamma -\mathrm{Fe}}$ are the charge density of corresponding isolated La_2_O_2_S slab and γ-Fe slab in the same interface structure, respectively.

Figure 5 shows the charge density differences around La(S)-terminated and S-terminated interface structures. There was an obvious charge transfer or redistribution in or near the interface region. By comparing the three-dimensional charge density difference of both interface structures, it was found that the electron accumulation was mainly concentrated below the La and S atoms, while the electron depletion was mostly on the upward side of the Fe atoms. The interface electron transfer was mainly contributed from the Fe–La and Fe–S atom pairs. Therefore, the interfacial bonds mainly came from the electronic interactions of the Fe–La and Fe–S atom pairs.

The localized distribution characteristics of electrons can be characterized by electron localization function (ELF), which can also confirm the bond types. The ELF was calculated by Equation (14) [45]:(14)$$ELF=1/\left[1+{\left(D_{r}/Dh_{r}\right)}^{2}\right],$$
where $D_{r}$ is the real electron gas density; and $Dh_{r}$ is the uniform electron gas density. The values of ELF are between 0 and 1. The upper limit ELF = 1 corresponds to perfect localization; ELF = 1/2 corresponds to electron gas-like pair probability. While the value ELF = 0 means that electron might be entirely delocalized (or that there are no electrons).

Figure 6 shows the ELF images of the La(S)-terminated and S-terminated interfaces when the isosurface level value was 0.1 eV/Bohr^3^. Chemical bonds were formed between the different types of interface atoms, and the charge was redistributed after the interface was formed. As shown in Figure 6a,b, the electronic spatial distribution in the atomic layer near the interface differed greatly from that in the innermost atomic layer. Moreover, the extranuclear electron orbitals of the La and S atoms had obvious interactions with that of the Fe atoms at the interface. It can be observed from Figure 6c that there were regions with ELF ≈ 0.3 at the interface. In this case, there was low electron localization, and an ionic bond was formed between the interfaces. Combined with Bader charge analysis, the electrons of La atoms at the interface were transferred to the Fe side, and a metal bond was formed between the interfaces. As shown in Figure 6d, there were green regions with ELF ≈ 0.5 between the S atoms and Fe atoms at the interface. Because of the strong electronegativity of S atoms, the electron loss was mostly concentrated on the Fe atoms at the S-terminated interface, while the electron accumulation was mainly on the S atoms, which contributed to the formation of covalent bonds at the S-terminated interface.

To further investigate the bonding characteristics, the partial density of states (PDOS) was also calculated, as shown in Figure 7. The dashed line indicates the position of the Fermi level. In addition, the DOS near both interfaces was not zero at the Fermi energy, indicating certain metallic characteristics. Obviously, the DOS of the La and Fe atoms at the first layer of the La(S)-terminated interface peaked at Fermi level. The metal characteristics of the La(S)-terminated interface were contributed by La atoms and Fe atoms, while those of the S-terminal interface were made by the Fe atomic contributions. As can be seen from Figure 7a, there was a weak electronic interaction between Fe and La atoms at the La(S)-terminated interface. By comparing the PDOS of different layers of Fe and La atoms, it was found that the d-orbital electrons of the subsurface and bulk La atoms had a more instinct localized feature than the interface La atoms. This suggests that the inner La atoms were more covalent. Moreover, several overlapped peaks appeared between the third layer of La atoms and the second layer of S atoms near the interface, indicating the existence of covalent bonds near the interface. It can be observed from Figure 7b that there was strong electron interaction between Fe atoms and S atoms at the S-terminated interface. In addition, there were some overlapped peaks between Fe-d orbital and the S-p orbital, forming a covalent bond between them at the S-terminated interface. Some overlapped peaks also appeared between the second layer of La atoms and the third layer of O atoms near the interface, which contributed to the formation of covalent bonds near the interface.


#### 3.3.4. Heterogeneous Nucleation Potency of γ-Fe on La_2_O_2_S Particles

To ensure sufficient atomic matching to retain a coherent or semi-coherent interface between the two phases, empirical criteria of the f_r_ and f_d_ values were both less than 10 pct in an as-cast light alloy based on the E2EM model [18,19,20,22]. According to the crystallographic calculation results in Table 3 and the predicted rough OR between La_2_O_2_S and γ-Fe, the minimum values of f_r_ and f_d_ in construction of the OR were 9.43% and 20.72%, respectively. This implies that, from a crystallographic point view, La_2_O_2_S is not an effective nucleant for the nucleation of γ-Fe.

Although all of the first-principles calculations were performed at 0 K, the results proved the experimental phenomena of solid-phase systems and solid–liquid interfaces at high temperatures [46,47]. To make La_2_O_2_S the effective heterogeneous nucleus of γ-Fe during the solidification, the solid–solid interface energy of La_2_O_2_S/γ-Fe should be between 0 J/m^2^ and 0.204 J/m^2^ from the viewpoint of thermodynamics [37], for which the theoretical value of liquid–solid interface energy is 0.241 J/m^2^ [48], as shown in Figure 4. As can be seen from Figure 4, the interface energy of La_2_O_2_S/γ-Fe could not meet the interface energy requirements as a heterogeneous nucleation interface. It did not comply with the thermodynamic conditions that make La_2_O_2_S particles effective as potent heterogeneous nucleation substrates for the γ-Fe matrix. In addition, factors, such as the size and distribution of La_2_O_2_S particles, also impact whether La_2_O_2_S can be an effective heterogeneous nucleus of γ-Fe during solidification. Future research will focus on the impact of such factors. The orientation relationship of crystallography will be verified by both electron backscatter diffraction and transmission electron microscopy.

## 4. Conclusions

With the crystallographic calculations using the E2EM model, a rough HCP/FCC orientation relationship (OR) was predicted between La_2_O_2_S particles and the γ-Fe matrix, the values of the interatomic spacing misfit and the interplanar spacing mismatch of which were 9.43% and 20.72%, respectively. This OR was ${\left[2\bar{1}\bar{1}0\right]}_{{\mathrm{La}}_{2}O_{2}S}$∥${\left[100\right]}_{\gamma -\mathrm{Fe}}$ and ${\left(000\bar{3}\right)}_{{\mathrm{La}}_{2}O_{2}S}$∥${\left(00\bar{2}\right)}_{\gamma -\mathrm{Fe}}$.For γ-Fe/La_2_O_2_S$\left(000\bar{3}\right)$ interfaces, the La(S)-terminated and S-terminated interfaces were more stable than others with smaller interface energy. In the range of $-5.12\;\mathrm{eV}\leq \Delta \mu_{\mathrm{La}}\leq -4.69\;\mathrm{eV}$, the S-terminated interface structure was the most stable. Nevertheless, for the range of $\Delta \mu_{\mathrm{La}}$ from −4.69 eV to Fermi energy level, the La(S)-terminated interface structure had the best stability. The S-terminated interface had the largest $W_{\mathrm{ad}}$, which was 4.403 J/m^2^, followed by the La(S)-terminated interface, which was 3.372 J/m^2^.According to the bonding analysis of the electronic structure and charge transfer on the two different interfaces, the interface interaction of the S-terminated interface exhibited metallic and ionic bonds, whereas the La(S)-terminated interface exhibited metallic and covalent bonds.Crystallographic calculations indicated that the interatomic spacing misfit and the interplanar spacing mismatch were large enough, compared with other reported efficient nucleating systems, leading to weak potency of La_2_O_2_S as a nucleation site for γ-Fe. The first-principles calculations revealed that the interfacial energy of La_2_O_2_S/γ-Fe cannot meet the interface energy requirements as a heterogeneous nucleation interface. Thus, La_2_O_2_S particles are noneffective heterogeneous nucleation substrates for a γ-Fe matrix from crystallography and thermodynamics viewpoints.

## Figures and Tables

**Figure 1 materials-15-01374-f001:**
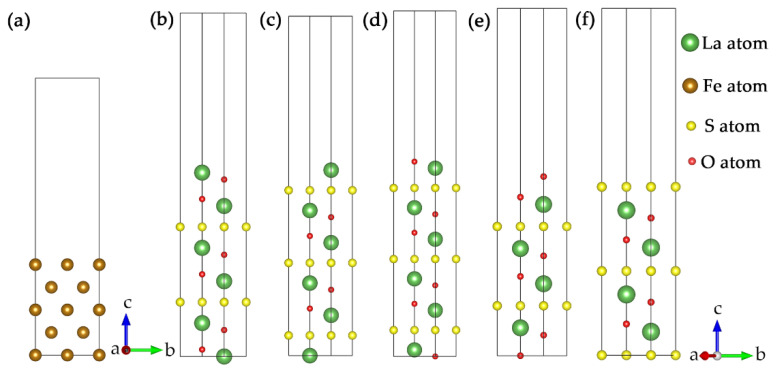
Slab configuration: (**a**) γ-Fe$\left(00\bar{2}\right)$ surface and La_2_O_2_S$\left(000\bar{3}\right)$ surfaces terminated with: (**b**) La(O)-terminated; (**c**) La(S)-terminated; (**d**) O(La)-terminated; (**e**) O(O)-terminated; (**f**) S-terminated. Coordinate dimension a, b and c represent *x*-axis, *y*-axis and *z*-axis respectively.

**Figure 2 materials-15-01374-f002:**
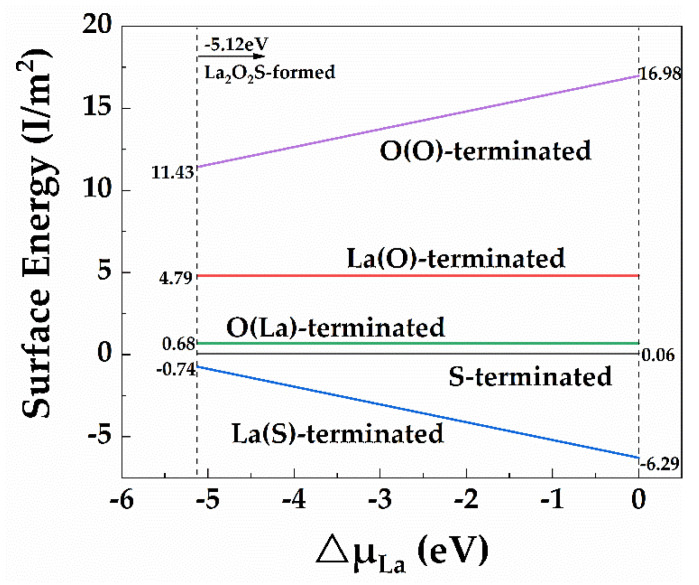
Surface energies for the La_2_O_2_S$\left(000\bar{3}\right)$ surfaces as $\Delta \mu_{\mathrm{La}}$. The corresponding number of atomic layers adopted in the surface configuration are 14, 13, 15, 12 and 11 for La(O)-terminated, La(S)-terminated, O(La)-terminated, O(O)-terminated and S-terminated surfaces, respectively.

**Figure 3 materials-15-01374-f003:**
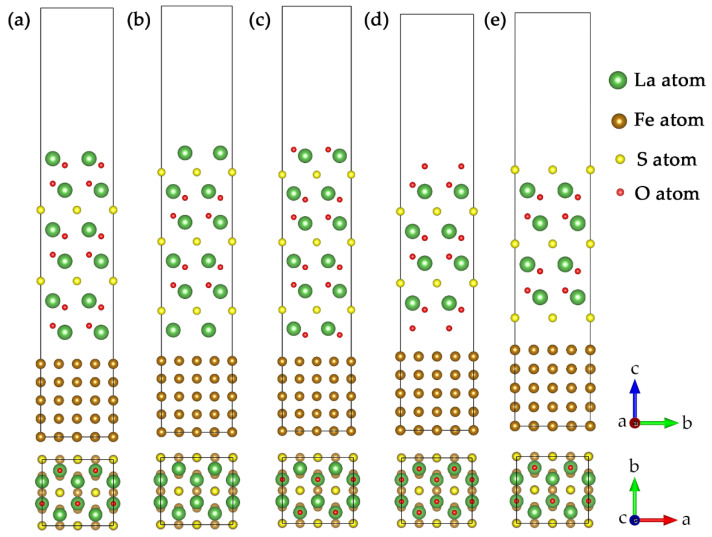
Supercell structures before relaxation: (**a**) La(O)-terminated La_2_O_2_S/γ-Fe interface; (**b**) La(S)-terminated La_2_O_2_S/γ-Fe interface; (**c**) O(La)-terminated La_2_O_2_S/γ-Fe interface; (**d**) O(O)-terminated La_2_O_2_S/γ-Fe interface; (**e**) S-terminated La_2_O_2_S/γ-Fe interface. Coordinate dimension a, b and c represent *x*-axis, *y*-axis and *z*-axis respectively.

**Figure 4 materials-15-01374-f004:**
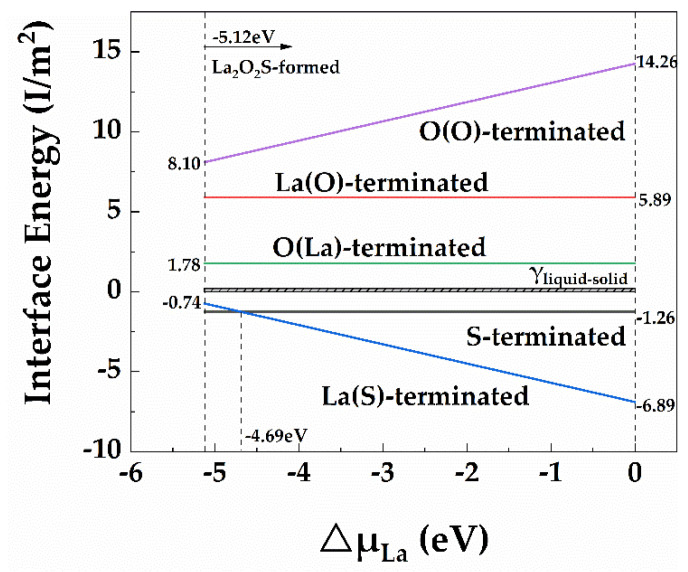
Interface energies for the La_2_O_2_S/γ-Fe interfaces with different terminated interfaces as $\Delta \mu_{\mathrm{La}}$.

**Figure 5 materials-15-01374-f005:**
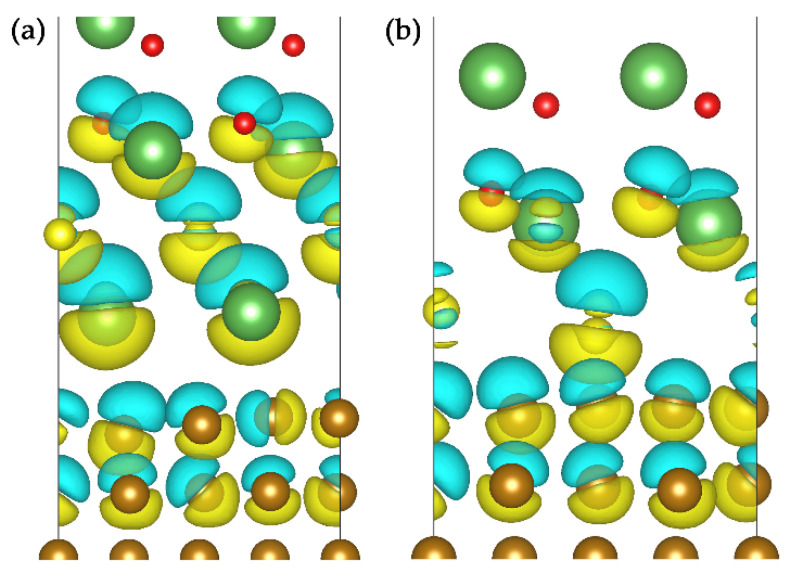
Three-dimensional charge density difference: (**a**) the La(S)-terminated La_2_O_2_S/γ-Fe interface; (**b**) S-terminated La_2_O_2_S/γ-Fe interface. The yellow regions indicate charge accumulation, and the blue regions represent charge loss.

**Figure 6 materials-15-01374-f006:**
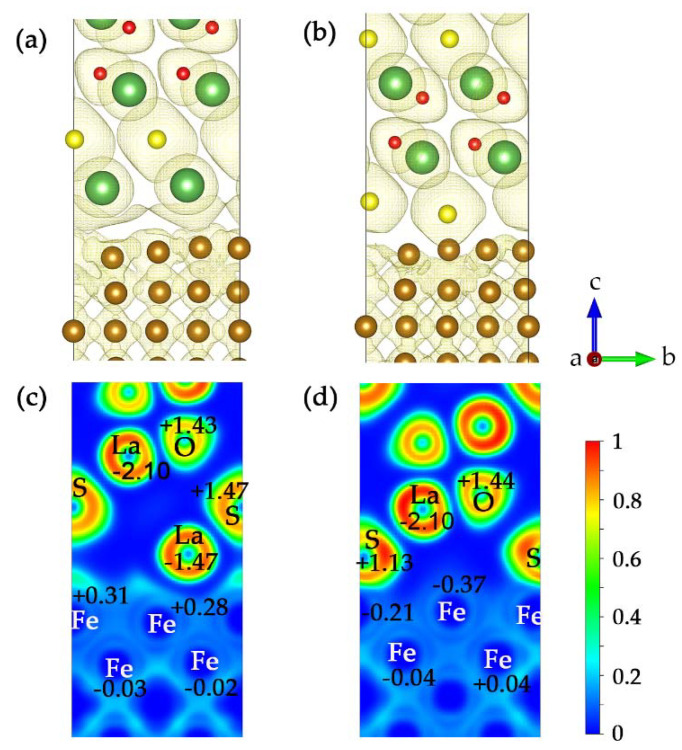
ELF isosurface and internal cross-section along plane (100) of γ-Fe/La_2_O_2_S interface: (**a**) ELF isosurface of the La(S)-terminated La_2_O_2_S/γ-Fe interface; (**b**) ELF isosurface of S-terminated La_2_O_2_S/γ-Fe interface; (**c**) ELF cross-section of S-terminated La_2_O_2_S/γ-Fe interface; (**d**) ELF cross-section of S-terminated La_2_O_2_S/γ-Fe interface (isosurface level = 0.1 eV/Bohr^3^). The numbers marked in the section denote the Bader charges (unit: e) of interfacial atoms. Positive and negative denote gain and loss of electrons respectively. Coordinate dimension a, b and c represent *x*-axis, *y*-axis and *z*-axis respectively.

**Figure 7 materials-15-01374-f007:**
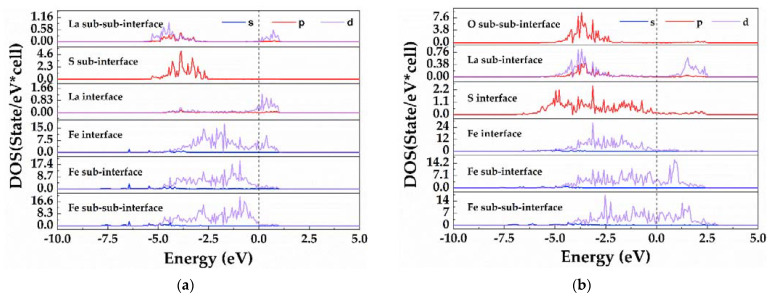
The partial density of states different atomic layer near the interface: (**a**) the La(S)-terminated La_2_O_2_S/γ- Fe interface; (**b**) S-terminated La_2_O_2_S/γ-Fe interface (the vertical dashed line indicates the position for Fermi level).

**Table 1 materials-15-01374-t001:** Crystal structure, space group, lattice parameters and atomic positions of γ-Fe and La_2_O_2_S.

Phase	Crystal Structure	Space Group	Lattice Parameters (nm)	Atomic Positions					
				Atom	Mult.	Wyck.	x	y	z
γ-Fe	FCC	$\mathrm{Fm}\bar{3}m$	a = 0.3654	Fe	4	a	0	0	0
La_2_O_2_S	Hexagonal	$P\bar{3}m1$	a = 0.4035 c = 0.6914	La	2	d	0.3333	0.6667	0.279
				O	2	d	0.3333	0.6667	0.629
				S	1	a	0	0	0

**Table 2 materials-15-01374-t002:** CP rows and CP planes of γ-Fe and La_2_O_2_S.

Phase	CP Rows	Interatomic Spacing (nm)	Type	CP Planes	Interplanes Spacing (nm)
γ-Fe	$\langle 2\bar{1}1\rangle$	0.224	ZZ	{111}	0.211
	$\langle 10\bar{1}\rangle$	0.258	SS	{020}	0.183
	〈100〉	0.365	SS	{022}	0.129
				{131}	0.110
La_2_O_2_S	〈0001〉	0.229	PS	$\left\{01\bar{1}1\right\}$	0.312
	$\langle \bar{1}101\rangle$	0.325	PS	$\left\{01\bar{1}2\right\}$	0.246
	$\langle 2\bar{1}\bar{1}0\rangle$	0.398	SS	{0003}	0.230
				$\left\{11\bar{2}0\right\}$	0.202

**Table 3 materials-15-01374-t003:** Interatomic spacing misfit (f_r_) along CP or nearly CP matching rows and interplanar spacing mismatch (f_d_) of CP or nearly CP matching planes between γ-Fe and La_2_O_2_S.

Matching Row Pairs	Misfit (f_r_) (%)	Matching Type	Matching Plane Pairs	Mismatch (f_d_) (%)
${\langle 2\bar{1}\bar{1}0\rangle }_{{\mathrm{La}}_{2}O_{2}S}$ $∥{\langle 100\rangle }_{\gamma -\mathrm{Fe}}$	9.43	SS-SS	${\left\{11\bar{2}0\right\}}_{{\mathrm{La}}_{2}O_{2}S}$ $∥{\left\{111\right\}}_{\gamma -\mathrm{Fe}}$	4.58
			${\left\{0003\right\}}_{{\mathrm{La}}_{2}O_{2}S}$ $∥{\left\{111\right\}}_{\gamma -\mathrm{Fe}}$	8.45
			${\left\{11\bar{2}0\right\}}_{{\mathrm{La}}_{2}O_{2}S}$ $∥{\left\{020\right\}}_{\gamma -\mathrm{Fe}}$	9.43
			${\left\{01\bar{1}2\right\}}_{{\mathrm{La}}_{2}O_{2}S}$ $∥{\left\{111\right\}}_{\gamma -\mathrm{Fe}}$	14.15
			${\left\{0003\right\}}_{{\mathrm{La}}_{2}O_{2}S}$ $∥{\left\{020\right\}}_{\gamma -\mathrm{Fe}}$	20.72
			${\left\{01\bar{1}2\right\}}_{{\mathrm{La}}_{2}O_{2}S}$ $∥{\left\{020\right\}}_{\gamma -\mathrm{Fe}}$	25.65

Note: ∥— Matching.

**Table 4 materials-15-01374-t004:** Calculated lattice constants, bulk module and equilibrium volume of the bulk γ-Fe compared with other references and experimental data.

Phase		Methods	a (nm)	B_0_ (GPa)	V_0_ (Å^3^/Cell)
γ-Fe	This work	LDA	0.3372	344	38.3
		GGA-PBE	0.3636	152	48.1
	Others	LDA [39]	0.3382	320	38.7
		GGA-PBE [25]	0.344	-	40.9
		GGA-PW91 [39]	0.3472	240	41.9
	Exp. [31]		0.3654	-	48.8

**Table 5 materials-15-01374-t005:** Calculated lattice constants, bulk module and equilibrium volume per atom of the bulk La2O2S compared with other references and experimental data.

Phase		Methods	a (nm)	c (nm)	c/a	B_0_ (GPa)	V_0_ (Å^3^/Cell)
La_2_O_2_S	This work	LDA	0.4065	0.6939	1.707	111	99.32
		GGA-PBE	0.4134	0.7073	1.711	102	104.67
	Others	LDA [40]	0.3982	0.6826	1.714	128	93.73
		GGA-PBE [40]	0.4055	0.6944	1.712	115	98.91
		GGA-PBE [41]	0.406	0.695	1.712	-	-
	Exp. [32]		0.4035	0.6914	1.713	-	97.49

**Table 6 materials-15-01374-t006:** Surface energy of γ-Fe$\left(00\bar{2}\right)$.

Layer	Surface Energy (J/m^2^)			
	5	7	9	11
γ-Fe$\left(00\bar{2}\right)$	2.18	2.18	2.19	2.20

**Table 7 materials-15-01374-t007:** Change of the interlayer distance of the La-terminated surface as a percentage of the respective bulk spacing.

Slab Thickness (n)								
d-Layer	La(O)-Terminated Surface				La(S)-Terminated Surface			
	9	14	19	24	8	13	18	23
d_1-2_	11.677	10.600	12.295	12.164	4.500	5.663	6.178	5.534
d_2-3_	−2.300	−1.508	−2.361	−2.443	−2.387	−2.121	−2.748	−1.732
d_3-4_		−0.475	−0.860	−0.774		0.184	−0.084	0.331
d_4-5_		−0.137	0.370	0.077		−0.773	−0.208	−0.788
d_5-6_			−0.454	−0.507			0.356	−0.323
d_6-7_				1.534			−0.079	−0.866
d_7-8_				0.466				−0.620

**Table 8 materials-15-01374-t008:** Change of the interlayer distance of the O-terminated surface as a percentage of the respective bulk spacing.

Slab Thickness (n)								
d-Layer	O(La)-Terminated Surface				O(O)-Terminated Surface			
	5	10	15	20	7	12	17	22
d_1-2_	8.076	8.532	8.116	8.426	−7.891	−9.082	−8.793	−9.164
d_2-3_	−0.899	−0.386	−0.410	−0.306	17.474	19.441	18.994	19.533
d_3-4_		−0.880	−0.576	−0.497		−1.594	−1.289	−1.489
d_4-5_			−0.221	−0.595		−0.192	−0.178	0.049
d_5-6_			−0.199	−0.039			−0.882	−0.280
d_6-7_				−0.043			−0.256	−0.286
d_7-8_				−0.010				0.424
d_8-9_								−0.004

**Table 9 materials-15-01374-t009:** Change of the interlayer distance of the S-terminated surface as a percentage of the respective bulk spacing.

Slab Thickness(n)				
d-Layer	S-Terminated Surface			
	6	11	16	21
d_1-2_	0.243	−1.037	−1.269	−0.929
d_2-3_	−1.093	−2.080	−1.917	−2.001
d_3-4_		−0.060	0.112	−0.089
d_4-5_			0.268	0.335
d_5-6_			−0.039	−0.768
d_6-7_				−0.466

**Table 10 materials-15-01374-t010:** Interface ideal adhesion energy $W_{\mathrm{ad}}$ for the five γ-Fe/La_2_O_2_S interface systems.

Interface	$W_{\mathrm{ad}}\;(J/m^{2})$
La(O)-terminated	2.887
La(S)-terminated	3.372
O(La)-terminated	2.132
O(O)-terminated	2.584
S-terminated	4.403

## Data Availability

The data presented in this study are available on request from the corresponding author.

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
