# Peer review of "Crystallographic Calculations and First-Principles Calculations of Heterogeneous Nucleation Potency of γ-Fe on La2O2S Particles"

_materials, 2022, doi:10.3390/ma15041374_

Round 1
Reviewer 1 Report
The manuscript with the title "Crystallographic Calculations and First-principles Calculations of Heterogeneous Nucleation Potency of γ-Fe on La2O2S Particles" is considered for Materials MDPI journal. The authors benchmarked the edge-to-edge matching model for the atomic matching mismatch and predicted the orientation relationship between La2O2S and γ−Fe based on the crystallographic point view.
The research is well organized, the manuscript is well written, contains a lot of details of the calculations and results. VASP with LDA/GGA+U is one of the best choices to investigate the La2O2S /γ-Fe interface. The Bader charges, electron localization function (ELF), along with the structural favorable orientation and PDOS studies, are suitable tools to investigate the electronic structure and bonding of the most stable La2O2S /γ-Fe interfaces.
There are some minor issues to be cleared in the manuscript:
1. The Introduction section contains no references for La2O2S as bulk or particles. This compound might be discussed in the references about La/Ce-containing inclusions, but it is not clear from the Introduction why this particular oxide is taken for the interface. The properties of La2O2S are not covered.
2. Line 134 is unfinished.
3. Magnetic properties and magnetic moments of Fe in the considered interface should be discussed. The spin-polarization calculations are done.
This work reports novel results of the first principles calculations for the Fe-La2O2S particles and warrant publication in Materials. There are a few minor corrections to be done before the publication.
Author Response
Point 1: The Introduction section contains no references for La2O2S as bulk or particles. This compound might be discussed in the references about La/Ce-containing inclusions, but it is not clear from the Introduction why this particular oxide is taken for the interface. The properties of La2O2S are not covered.
Response 1: Thank you very much for the reviewer’s professional comments on the article. According to your suggestions, we re-organized the section of introduction. We added some references about the solidification refinement of the γ-austenite in steels through lanthanum addition and the lattice disregistry between La2O2S and γ-austenite. The addition of lanthanum can refine the solidification structure, but not all lanthanum inclusions can be acted as heterogeneous nucleus of γ-austenite. Hence, it is necessary to clarify the heterogeneous nucleation potency of La-containing inclusion phases for the primary γ-austenite. These papers can make the introduction of this paper more convincing and comprehensive. The added portions in the introduction and references were highlighted in red. (Page 2, line 58-67, 81-82).
Point 2: Line 134 is unfinished.
Response 2: Thanks to the reviewer for his professional comments on the article. We have corrected the grammar errors of the article, and has made corresponding deletion in the revised draft.
Point 3: Magnetic properties and magnetic moments of Fe in the considered interface should be discussed. The spin-polarization calculations are done.
Response 3: Thank you very much for the reviewer’s professional comments on the article. To obtain more accurate interface characteristics, we have employed the spin-polarization method in the DFT calculations for La2O2S and γ-austenite system. In this work, we mainly focus on the discussion of La2O2S and γ-austenite interface stability. Properties such as interfacial magnetism will be further investigated in the subsequent research work.

Reviewer 2 Report
In this work authors present a rather detailed investigation by first-principles calculations of both gamma-iron and La2O2S. The ultimate goal is to understand if La2O2S nanoparticles can play an important role in influencing nucleation of gamma-iron. The work includes first a simple screening, based on the so called E2EM method, of the various potential interfaces between Fe and La2O2S, followed by first-principle calculations of surface and interface energies for the selected relative orientations. The final outcome is that, mainly due to high interface energies, LA2O2S are not effective substrates for triggering heterogeneous nucleation of gamma-Fe. The topics is of interest, but there are a couple of issue that authors should fix prior to publication
Minor points:
1) The ab initio approach per se looks solid: authors use VASP and LDA or GGA/PBE functionals + corrections when needed. Also, they did check for convergence of results in terms of the number of layers in their slab.
The only parameter which does not fully convince me is the criterium for declaring converged structure optimizations: 0.03eV/A is still a rather large force. Did author check for a couple of targetted cases that result do not change significantly using a stricter criterium (e.g. 0.005eV/A)?
2) I believe that lines 95, 96, 97 (page 3) were left in the text unintentionally.
3) A more significant point is the following one: the final result obtained by the authors is that the investigated system is NOT a good one for triggering nucleation. Well, this makes the full paper a little questionable: what is the message? Not to use La2O2S? Were people using it? What I mean is that the full introduction and motivation of the paper should be re-thought in view of the final negative result. Authors should, e.g. motivate the study of the various interface energies even if they turn out to be high and as such not suitable for the application which authors had in mind, otherwise it sounds awkward.
Author Response
Point 1: The ab initio approach per se looks solid: authors use VASP and LDA or GGA/PBE functionals + corrections when needed. Also, they did check for convergence of results in terms of the number of layers in their slab.
The only parameter which does not fully convince me is the criterium for declaring converged structure optimizations: 0.03eV/A is still a rather large force. Did author check for a couple of targetted cases that result do not change significantly using a stricter criterium (e.g. 0.005eV/A)?
Response 1: Thanks to the reviewer for his professional comments on the article. A smaller force convergence criterion can indeed simulate the interface more accurately when studying the interface system, but it also means that more computational power will be required. In previous research works (Refs. [27-29]), the Hellmann-Feynman force convergence criterion is generally adopted as 0.03eV/Å considering the computational workload. The computation methodology and parameters are validated to provide sufficient precision in present calculations and references to relevant literature have been added in the corresponding positions in the article. (Page 3, line 133).
Point 2: I believe that lines 95, 96, 97 (page 3) were left in the text unintentionally.
Response 2: Thanks to the reviewer for his careful comments on the article. We have made corresponding deletion in the revised draft. We are very apologetic about the mistake due to our negligence.
Point 3: A more significant point is the following one: the final result obtained by the authors is that the investigated system is NOT a good one for triggering nucleation. Well, this makes the full paper a little questionable: what is the message? Not to use La2O2S? Were people using it? What I mean is that the full introduction and motivation of the paper should be re-thought in view of the final negative result. Authors should, e.g. motivate the study of the various interface energies even if they turn out to be high and as such not suitable for the application which authors had in mind, otherwise it sounds awkward.
Response 3: Thank you very much for the reviewer’s professional comments on the article. According to your suggestions, we re-thought the motivation and re-organized the section of introduction. We added some references about the solidification refinement of the γ-austenite in steels through lanthanum addition and the lattice disregistry calculation between La2O2S and γ-austenite. Previous studies have shown that the addition of lanthanum can refine the solidification structure, and the solidification structure refinement mechanism is mainly the heterogeneous nucleation effect of the primary γ-austenite on La-containing inclusions with high melting points, including oxides, oxysulfides and sulfides. However, not all lanthanum inclusions can be acted as heterogeneous nucleus of γ-austenite. It is not specified which La-inclusions serve as heterogeneous nucleation sites. Hence, it is necessary to clarify the heterogeneous nucleation potency of La-containing inclusion phases for the primary γ-austenite. In this work, La2O2S, which is the typical oxysulfide was selected. We aims to clarify the heterogeneous nucleation potency of γ-Fe on La2O2S particles combining a crystallographic calculation using the edge-to-edge matching model with the first-principles calculations from the energetics point of view. The added portions and references in the introduction were highlighted in red. (Page 2, line 58-67, 81-82).

Round 2
Reviewer 2 Report
Authors correctly answered to my previous concerns and re-wrote part of the introduction to better motivate the paper that I now recommend for publication.